# Information Geometry of the Retinal Representation Manifold

**Xuehao Ding**[1], **Dongsoo Lee**[2], **Joshua B. Melander**[3], **George Sivulka**[4], **Surya Ganguli**[5], and **Stephen A. Baccus**[6]

[1,5]Department of Applied Physics, Stanford University
[2,3]Neurosciences Phd Program, Stanford University
[4]Department of Electrical Engineering, Stanford University
[6]Department of Neurobiology, Stanford University
*{xhding, dsnl, melander, gsivulka, sganguli, baccus}@stanford.edu*

## Abstract

The ability for the brain to discriminate among visual stimuli is constrained by their retinal representations. Previous studies of visual discriminability have been limited to either low-dimensional artificial stimuli or pure theoretical considerations without a realistic encoding model. Here we propose a novel framework for understanding stimulus discriminability achieved by retinal representations of naturalistic stimuli with the method of information geometry. To model the joint probability distribution of neural responses conditioned on the stimulus, we created a stochastic encoding model of a population of salamander retinal ganglion cells based on a three-layer convolutional neural network model. This model not only accurately captured the mean response to natural scenes but also a variety of second-order statistics. With the model and the proposed theory, we computed the Fisher information metric over stimuli to study the most discriminable stimulus directions. We found that the most discriminable stimulus varied substantially across stimuli, allowing an examination of the relationship between the most discriminable stimulus and the current stimulus. By examining responses generated by the most discriminable stimuli we further found that the most discriminative response mode is often aligned with the most stochastic mode. This finding carries the important implication that under natural scenes, retinal noise correlations are information-limiting rather than increasing information transmission as has been previously speculated. We additionally observed that sensitivity saturates less in the population than for single cells and that as a function of firing rate, Fisher information varies less than sensitivity. We conclude that under natural scenes, population coding benefits from complementary coding and helps to equalize the information carried by different firing rates, which may facilitate decoding of the stimulus under principles of information maximization.

## 1 Introduction

Neural populations represent information with their collective activity. In the field of sensory neuroscience, whereas there is much focus on characterizing the sensitivity of neurons to different stimuli, the true function of a sensory system is to discriminate between stimuli. A quantitative description of discriminability in a neural population requires not only a precise knowledge of sensitivity to stimuli, but also the structure of noise correlations in the population.

37th Conference on Neural Information Processing Systems (NeurIPS 2023).

Neural representation manifolds constitute a modern approach to studying neural populations in a geometric framework, and recent machine learning literature has seen extensive research on the geometry of representation manifolds of neural networks trained for artificial tasks [1, 2, 3, 4, 5]. In separate work, Fisher information has been widely utilized in neuroscience for measuring decoding fidelity, particularly in theoretical studies exploring the effect of noise correlations [6, 7, 8]. In this study, we propose a framework for studying the Fisher information geometry of representation manifolds for natural visual scenes in a population of visual neurons, in particular the retina. In addition, for the first time, we apply such a manifold analysis to the neural representation of natural stimuli for real data.

The discriminability of biological visual systems has long been studied for low-dimensional and discrete artificial stimuli [9, 10, 11]. However, naturalistic visual stimuli in the real world are inherently high-dimensional and discriminability is generally heterogeneous across different types of changes in a stimulus. Recent advances in convolutional neural networks (CNNs) have enabled the accurate prediction of retinal responses to naturalistic stimuli, as well as ethological computations [12, 13, 14]. Nevertheless, previous work has predominantly focused on deterministic trial-averaged neural codes, which is not adequate for computing discriminability. Thus an encoding model of stochastic retinal representations for natural scenes is required, which constitutes one of the primary contributions of this work.

We present here a highly accurate model of both sensitivity and stochasticity for the retinal ganglion cell (RGC) population, thus creating the first accurate model of discriminability in a neural population for its natural inputs. From an analysis of the Fisher information geometry of the retinal representation of stimuli, we find that the most discriminable direction for the RGC population varies greatly across stimuli and is correlated with the mean neural response. We further address a long standing question as to the effect of noise correlations on information. We find that the most discriminative mode in the representation space often aligns with the most stochastic direction, implying that noise correlations in the retina are detrimental to information coding. Finally, we analyze how discriminability depends on firing rate of individual cells and in the population, and discuss the implications in relation to principles of information maximization.

## 2   Theory

In this section, we describe basic theory of information geometry [15] and its application to sensory neuroscience. Let $\boldsymbol{x}$ denote the stimulus vector and $\boldsymbol{y}$ denote the spike count vector whose dimension is equal to the number of neurons encoding the stimulus. Each stimulus induces a conditional probability distribution of sensory neural responses $P(\boldsymbol{y}|\boldsymbol{x})$ and mean firing rates $\boldsymbol{\mu}(\boldsymbol{x}) := E[\boldsymbol{y}|\boldsymbol{x}]$. Along any decoding vector $\boldsymbol{a}$, the sensitivity is defined as $||\nabla_{\boldsymbol{x}} \langle \boldsymbol{\mu}(\boldsymbol{x}), \boldsymbol{a} \rangle ||$, while the stochasticity is defined as the standard deviation of $\langle \boldsymbol{y}, \boldsymbol{a} \rangle$ conditioned on $\boldsymbol{x}$ (Figure 1). In our case in the retina, $\boldsymbol{\mu}(\boldsymbol{x})$ is modeled by a noiseless CNN and $P(\boldsymbol{y}|\boldsymbol{x})$ is modeled by the stochastic counterpart.

The representation manifold is defined as the manifold of mean neural responses induced by stimuli: $\{\boldsymbol{\mu}(\boldsymbol{x})\}$. We also define a statistical manifold of conditional response distributions $\{P(\boldsymbol{y}|\boldsymbol{x})\}$ parameterized by stimuli $\boldsymbol{x}$, which yield a natural coordinate system. To equip the statistical manifold with a Riemannian metric, we consider the Kullback–Leibler divergence between two nearby distributions:

$$
\begin{aligned}
ds^2(\boldsymbol{x}, \boldsymbol{x} + d\boldsymbol{x}) &:= 2 \cdot D_{KL}[P(\boldsymbol{y}|\boldsymbol{x})||P(\boldsymbol{y}|\boldsymbol{x} + d\boldsymbol{x})] \\
&\approx -2 \int P(\boldsymbol{y}|\boldsymbol{x}) \ln \left[ 1 + \frac{dP(\boldsymbol{y}|\boldsymbol{x})}{P(\boldsymbol{y}|\boldsymbol{x})} \right] d\boldsymbol{y} \\
&\approx \sum_{ij} \int P(\boldsymbol{y}|\boldsymbol{x}) \frac{\partial \ln P(\boldsymbol{y}|\boldsymbol{x})}{\partial \boldsymbol{x_i}} \frac{\partial \ln P(\boldsymbol{y}|\boldsymbol{x})}{\partial \boldsymbol{x_j}} d\boldsymbol{y} d\boldsymbol{x_i} d\boldsymbol{x_j} \\
&= d\boldsymbol{x}^T \cdot \boldsymbol{I}(\boldsymbol{x}) \cdot d\boldsymbol{x},
\end{aligned}
\tag{1}
$$

where $\boldsymbol{I}(\boldsymbol{x})$ is the Fisher information matrix with respect to the stimulus, and the KL divergence is linearized up to the second order. The Fisher information metric $g^{stim}|_{\boldsymbol{x}}(\boldsymbol{e_i}, \boldsymbol{e_j}) = \boldsymbol{I}(\boldsymbol{x})_{ij}$ naturally characterizes the local discriminability of stimuli near $\boldsymbol{x}$.

When we assume that neural responses follow multivariate Gaussian distributions $\boldsymbol{y}|\boldsymbol{x} \sim N(\boldsymbol{\mu}(\boldsymbol{x}), \boldsymbol{\Sigma}(\boldsymbol{x}))$ and the covariance matrix is locally independent of the stimulus, $\frac{d\boldsymbol{\Sigma}(\boldsymbol{x})}{d\boldsymbol{x}} = \boldsymbol{0}$,

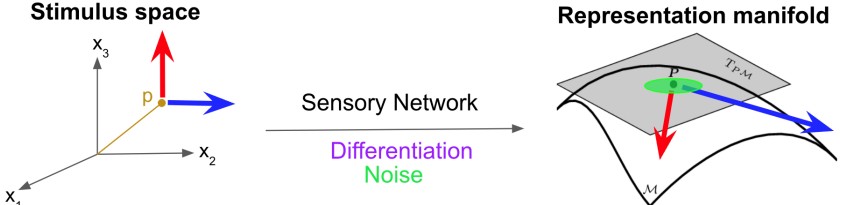

Figure 1: Schematic of basic concepts in a sensory system. The sensory network maps from the stimulus space to the representation manifold. The differentiation operator maps the stimulus tangent space to the representation tangent space. With noise in the network, each stimulus corresponds to a cloud of points in the representation space. Generally, the sensitivity (the blue and the red arrows) and the stochasticity (the green oval) are both heterogeneous.

the Fisher information metric can be expressed as

$$\boldsymbol{I}(\boldsymbol{x}) = \frac{d\boldsymbol{\mu}(\boldsymbol{x})}{d\boldsymbol{x}}^T \cdot \boldsymbol{\Sigma}^{-1}(\boldsymbol{x}) \cdot \frac{d\boldsymbol{\mu}(\boldsymbol{x})}{d\boldsymbol{x}}, \tag{2}$$

where $\frac{d\boldsymbol{\mu}(\boldsymbol{x})}{d\boldsymbol{x}}$ is the Jacobian matrix. Eq. 2 is commonly termed linear Fisher information which bounds the performance of unbiased linear decoders, which may be more biologically plausible than nonlinear decoders [16, 17, 18]. The latter assumption is a good approximation in our case because a neural network consisting only of ReLU nonlinearities is locally equivalent to an effective linear network [19] and the covariance matrix of a linear network is a constant matrix. This metric expression is interpretable in that discriminability is proportional to sensitivity and inversely proportional to noise. The length of a geodesic connecting two stimuli under this information metric measures the discriminability between them, when represented by the neural population.

Equivalently, the mean response $\boldsymbol{\mu}(\boldsymbol{x})$ on the representation manifold can also serve as a coordinate system. Then the metric is given by $g^{resp}|_{\boldsymbol{x}}(\boldsymbol{e_i}, \boldsymbol{e_j}) = \boldsymbol{\Sigma}^{-1}(\boldsymbol{x})_{ij}$. The metric matrix transforms between the stimulus space and the representation space as a tensor [1].

We define here several important quantities for analysis. The eigenvector of $\boldsymbol{I}(\boldsymbol{x})$ associated with the largest eigenvalue is the most discriminable direction in the stimulus tangent space, which we term the most discriminable input (MDI) at stimulus $\boldsymbol{x}$. The square root of the eigenvalue is the corresponding discriminability [20]. The MDI in the stimulus space can be pushed forward to the most discriminative response mode (MDR) in the representation tangent space via the Jacobian matrix: MDR $\propto \frac{d\boldsymbol{\mu}(\boldsymbol{x})}{d\boldsymbol{x}} \cdot$ MDI. Similarly, the most sensitive direction in the stimulus (input) tangent space (MSI) can be obtained as the principal eigenvector of $\frac{d\boldsymbol{\mu}(\boldsymbol{x})}{d\boldsymbol{x}}^T \cdot \frac{d\boldsymbol{\mu}(\boldsymbol{x})}{d\boldsymbol{x}}$ and the associated most sensitive response mode (MSR) obeys MSR $\propto \frac{d\boldsymbol{\mu}(\boldsymbol{x})}{d\boldsymbol{x}} \cdot$ MSI. Additionally, the most stochastic (noisy) response mode (MNR) is defined as the principal eigenvector of the noise covariance matrix $\boldsymbol{\Sigma}(\boldsymbol{x})$, which is no more than principal component analysis. For a ReLU neural network with small stochasticity, the stimulus dependence of the aforementioned tangent space features is solely derived from the stimulus dependence of the effective linear network [19].

One practical challenge of the above analysis with natural visual stimuli is the extremely high dimensionality of the stimulus space. Fortunately, we only need to compute the Fisher information matrix in a subspace defined by the recorded cell population. First we define the instantaneous receptive field for each cell as the direction of the greatest sensitivity to the stimulus at one stimulus point, which is the gradient of that cell's response, $\frac{d\boldsymbol{\mu}_i(\boldsymbol{x})}{d\boldsymbol{x}}$ [13]. Then we only need to compute Fisher information in the subspace spanned by instantaneous receptive fields, as justified by the following theorem.

**Theorem**: The top $N$ most discriminable directions are linear combinations of the instantaneous receptive fields, where $N$ is the number of output neurons.

**Proof:** Let $IR := \text{span}(\{\frac{d\boldsymbol{\mu}_i(\boldsymbol{x})}{d\boldsymbol{x}}\})$ represent the subspace spanned by instantaneous receptive fields. Suppose that the most discriminable direction $\boldsymbol{v_1}$ is not in $IR$, then we can decompose $\boldsymbol{v_1} = \boldsymbol{v_1^o} + \alpha\boldsymbol{v_1^p}$, where $\boldsymbol{v_1^o}$ is the orthogonal component in $IR^\perp$ and $\alpha\boldsymbol{v_1}^p$ is the parallel component

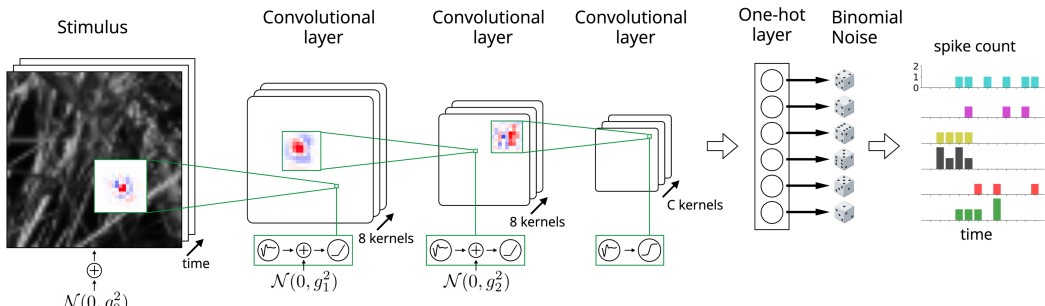

Figure 2: Model architecture. The first layer consists of eight $15 \times 15$ spatiotemporal convolutional filters. The second layer consists of eight $11 \times 11$ filters. The final layer consists of $C$ $9 \times 9$ filters, where $C$ represents the number of cell types. Batch normalization is applied after each convolutional filter. The nonlinearities in the first two layers are ReLUs, while in the last layer, the nonlinearity consists of the SoftPlus function followed by a parametric $\tanh$ function. Gaussian white noise is added to the stimulus and the first two convolutional layers, and independent binomial noise is applied to the final nonlinearity. The one-hot layer selects the location of the unit in the final convolutional layer to match that of the recorded neuron. Further details are explained in the main text.

in $IR$, $||\boldsymbol{v_1}|| = ||\boldsymbol{v_1^p}|| = 1$, $|\alpha| < 1$. Then following Eq. 2 we have

$$
\begin{aligned}
g^{stim}(\boldsymbol{v_1}, \boldsymbol{v_1}) &= \alpha^2 g^{stim}(\boldsymbol{v_1^p}, \boldsymbol{v_1^p}) \\
&< g^{stim}(\boldsymbol{v_1^p}, \boldsymbol{v_1^p}),
\end{aligned}
\tag{3}
$$

which contradicts with the fact that $\boldsymbol{v_1}$ is the most discriminable direction. Therefore, $\boldsymbol{v_1}$ is in $IR$.

Replacing $IR$ with $IR \cap \{\boldsymbol{v_1}\}^\perp$, with the same analysis we can prove that the second most discriminable direction $\boldsymbol{v_2}$ is also in $IR$. Subsequently, the whole theorem can be proven.

## 3 Methods

The spiking activity of a population of tiger salamander retinal ganglion cells was recorded in response to a sequence of jittering natural images. We created a stochastic encoding model capturing $P(\boldsymbol{y}|\boldsymbol{x})$ with the CNN architecture that included independent noise in the stimulus and in each unit of the CNN. The CNN parameters were trained using backpropagation to fit the mean firing rates $\boldsymbol{\mu}(\boldsymbol{x})$ to create a deterministic model, and then these parameters were frozen. Noise parameters included one parameter for each layer to set the standard deviation of the independent noise, and one parameter for binomial noise in each ganglion cell. These were then optimized by maximizing the log-likelihood of the empirical data and exploring the parameter space to match a few second-order statistics of the neural responses. Once the parameter optimization was completed, we computed the Fisher information metric with the model numerically for each stimulus and analyzed the most discriminable directions.

### 3.1 Stochastic model

The model architecture is shown in Figure 2. The model takes a spatiotemporal visual stimulus as an input and its output is a set of spike counts, one for each neuron in each time bin. The deterministic part of the model including its hyperparameters and training method is adopted from Ref. [12, 13, 14] which is a three-layer CNN chosen after an extensive architecture search. This CNN is the SOTA retinal encoding model for natural scenes in that (1) the prediction accuracy of RGC mean responses outperforms other models; (2) the activity of internal units is correlated with retinal interneuron responses recorded separately and not used in training; and (3) the model trained only on natural scenes can reproduce a wide range of phenomena induced by artificial stimuli. Thus, the deterministic part of our model has a clear correspondence with the real retina in that its circuitry is mechanistically interpretable and it captures many retinal computations. Every convolutional filter in our model was implemented by linearly stacking a sequence of $3 \times 3$ small filters, which outperforms the traditional method [21]. A parametric $\tanh$ nonlinearity was attached to the last convolutional layer for the

purpose of enforcing the refractory period constraint:

$$f(x; r_{max}, a, b) = \frac{r_{max}}{2} \cdot [\tanh(a(x-b)) + 1],  \tag{4}$$

where $a, b$ are cell-dependent model parameters, and $r_{max}$ represents the cell-dependent maximal firing rate.

The essential module that enables us to study retinal population coding is the one-hot layer. Model parameters are fit with the one-hot layer, which bijectively maps a few output units of the final convolutional layer to recorded neurons by the end of the training. The one-hot layer is then removed after parameter optimization so that we can analyze the representation of the full population, typically consisting of thousands of units. The rationale behind this method is the mosaic organization of each retinal cell type, in which all ganglion cells of a given type tile the retina with their dendrites [11]. Formally, the one-hot layer is defined as

$$\text{out}(k) = \sum_{ij} w_{ij}^k \cdot \text{input}(C(k), i, j), \ \forall \, k$$
$$\sum_{ij} w_{ij}^k = 1, \ \forall \, k  \tag{5}$$
$$w_{ij}^k \geq 0, \ \forall \, k, i, j$$

where the input is the output of the last convolutional layer, $k$ is the index of recorded neurons, $i, j$ (and $p, q$ below) are location indices, $w^k$ is the linear combination weight vector, and $C(k)$ is the neuron-to-channel map. Each $w^k$ will converge to a one-hot array after model training with our one-hot loss function, which is adapted from the semantic loss in Ref. [22]

$$\mathcal{L}_{\text{one-hot}}(w^k) \propto -\ln \sum_{pq} | \prod_{ij} (w_{ij}^k + \delta_{pq, ij} - 1)|,$$
$$\delta_{pq, ij} = \begin{cases} 1 & p = i, q = j \\ 0 & \text{otherwise} \end{cases},  \tag{6}$$

and which reflects the locations of recorded neurons. Each output channel represents a ganglion cell type and $C(k)$ was determined using hierarchical clustering of output channels (see the Supplementary Material for details).

Gaussian white noise was added to the stimulus to simulate stochasticity in photoreceptors, and was also added to pre-threshold signals in the first two layers to simulate stochasticity in bipolar cells and amacrine cells. We found that Gaussian noise outperforms Poisson noise in these layers, which can presumably be attributed to the noise generation mechanism in the retina and the central limit theorem. The stochasticity of ganglion cells was modeled as a parametric binomial probability mass function:

$$p^B(n; r, k, N) \propto \frac{\Gamma(k \cdot N + 1)}{\Gamma(k \cdot (N - n) + 1) \cdot \Gamma(k \cdot n + 1)} r^{k \cdot n} \cdot (1 - r)^{k \cdot (N - n)}, \quad n = 0, 1, ..., N  \tag{7}$$

where $n$ denotes the spike count in a time bin, $r$ is the rate parameter having a one-to-one correspondence with the mean firing rate $\mu$, $k$ is the variability parameter to be optimized, and $N$ is the cell-dependent upper bound of spike counts in one time bin reflecting the refractory period of that cell, which one can directly determine from data. In practice, for given $k$ and $N$ we can interpolate the $r - \mu$ mapping and use it to determine the rate parameter producing the desired mean firing rate, i.e., $\mu(r) = E_{p(n; r, k, N)}(n)$. Although we have explored a variety of probability mass functions to fit the sub-Poisson statistics of retinal ganglion cells [23, 24], the binomial noise Eq. 7 performs the best among them (see the Supplementary Material) and also is the most natural choice considering the refractory period constraint. We stress that the binomial noise on ganglion cells is independent noise, and thus will not qualitatively affect the geometry and the functional implication of noise correlations.

## 3.2 Optimization

Since the second order statistics are functions of the entire dataset and cannot be decomposed into terms depending on the individual stimulus, we optimized deterministic parameters and stochastic

parameters separately. CNN parameters and one-hot parameters $\{w^k\}$ were optimized together with the standard Poisson loss function and $\mathcal{L}_{\text{one-hot}}$ to fit mean firing rates smoothed using a $10\ ms$ Gaussian filter. The optimization was performed using ADAM [25] via Pytorch [26] on NVIDIA TITAN Xp, GeForce GTX TITAN X, GeForce RTX 3090, TITAN RTX, TITAN V, and GeForce RTX 2080 Ti GPUs. The network was regularized with an L2 weight penalty at each layer and an L1 penalty on the model output. Then $\texttt{tanh}$ parameters of the final nonlinearity were fitted with the least square regression. Then we optimized stochastic parameters while freezing all deterministic parameters. Variability parameters $k$ were optimized by maximizing the log-likelihood of recorded spike counts based on empirical firing rates. Standard deviations of Gaussian noise $\{g_0, g_1, g_2\}$ in each layer were optimized through grid searches in order to best fit empirical noise correlations and stimulus correlations by minimizing the smoothed and weighted average of mean squared errors.

### 3.3 Computation of Fisher information metric

For each stimulus, the noise covariance matrix $\boldsymbol{\Sigma}$ was estimated by repeatedly simulating the stochastic model thousands of times. We then computed the eigendecomposition of $\boldsymbol{\Sigma}$ obtaining local information of the representation manifold: $\boldsymbol{\Sigma} = \sum_i \lambda_i \boldsymbol{v_i} \boldsymbol{v_i}^T$. We applied the Gram–Schmidt orthogonalization to instantaneous receptive fields to obtain a basis of $IR$, which is a subspace of the stimulus tangent space. Then following Eq. 2, the Fisher information matrix with respect to the stimulus can be computed as

$$\boldsymbol{I} = \sum_i \frac{d\,\langle \boldsymbol{\mu}(\boldsymbol{z}), \boldsymbol{v_i}\rangle}{d\boldsymbol{z}}^T \lambda_i^{-1} \frac{d\,\langle \boldsymbol{\mu}(\boldsymbol{z}), \boldsymbol{v_i}\rangle}{d\boldsymbol{z}}, \tag{8}$$

where $\boldsymbol{z}$ is the stimulus coordinate under the $IR$ basis. In practice, we summed only over hundreds of the most stochastic modes that can explain $85\%$ to $90\%$ of the variance, as the sensitivity and stochasticity of the other modes are both close to zero indicating that those modes are orthogonal to the tangent space of the representation manifold. The gradient with respect to the stimulus was computed using the noiseless model.

## 4 Results

### 4.1 Model evaluation

Early stochastic encoding models of the retina were primarily developed for artificial stimuli [27, 28, 29], and their results cannot be directly generalized to natural scenes due to the stimulus-dependent nature of noise in the retina. Here our fitted model was evaluated by comparing with neural data various commonly used second-order statistics including correlation measures and single-cell variability measures (Figure 3). Surprisingly, the model prediction matches the neural data remarkably well despite that the model only has a few noise parameters. This result suggests that because noise in the retina propagates through the same network pathways as the signal, they are both captured by the CNN structure. This indicates a strikingly simple origin for noise correlations between all ganglion cell pairs: independent noise that starts in individual upstream cells, and then propagates through the same network that creates mean stimulus sensitivity. Statistics for preparations with longer test sequences tend to be more reliable and consequently our model performs better on these. For the first order statistics (mean firing rates), the Pearson correlation coefficient between model predictions and data ranges from $70\%$ to $80\%$, which is similar to Ref. [12, 13]. Such an accurate model of both sensitivity and stochasticity guarantees that the computation of Fisher information matrix is reliable (Eq. 2).

### 4.2 The most discriminable directions

Some examples of the most discriminable stimulus direction (MDI) and the most discriminative response mode (MDR) are shown in Figure 4a (see the Supplementary Material for more examples). Often the top eigenvalues were similar, and so to avoid over-interpreting the importance of individual eigenmodes, we found the sparsest direction (with the least L1 norm) in the space spanned by the top 5 MDIs for the purpose of visualization. Only spatial components are presented here as we found that temporal components for different stimuli are both very similar to each other and to those of instantaneous receptive fields [13]. We found that the MDI is strongly stimulus dependent, and is

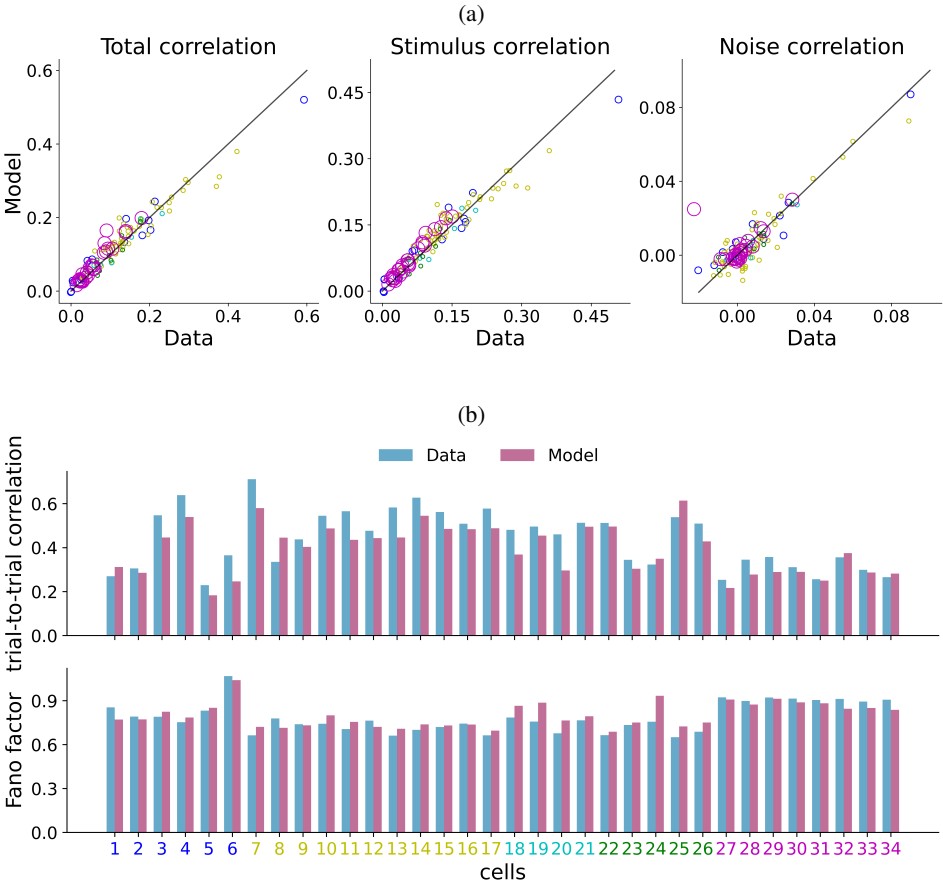

Figure 3: The model captures a variety of second-order statistics of neural responses (see the Supplementary Material for definitions). (a) Comparison of total pairwise correlation, stimulus correlation, and noise correlation between neural data and model prediction. Black lines represent perfect agreement. Each circle corresponds to a pair of neurons in an experimental preparation coded by its color and the circle radius is proportional to the square root of the total length of the test set. (b) Comparison of single-cell variability measures including the Fano factor and trial-to-trial correlation between neural data and model prediction. Neuron indices share the same color code with (a).

typically localized in a region where the sensitivity is high, an effect that arises from the locality of convolutional connections. The MDI tends to appear around the central region (red box in Figure 4a) because pixels within this area affect more output units than pixels near the edges or corners.

One general observation is that the MDI varies greatly across stimuli. From a mathematical point of view, this is to say that certain features of the tangent space depend on the base point. With respect to neural function, this variability potentially indicates the presence of adaptation to the stimulus. We found that the most discriminable direction is related to the base point through the representation space such that the MDR and the mean response are correlated (Figure 4b). This correlation can be attributed to the fact that neurons with intermediate firing rates exhibit the highest single-cell sensitivity and discriminability (Figure 6a). As a result, the MDI sometimes captures the most salient feature of the stimulus (the first row of Figure 4a), although this is not always the case (the second row of Figure 4a). Similar stimuli and mean responses likely produce distinct MDIs especially when their mean response pattern has multiple spatial components that can each yield a localized MDR (the third and fourth rows of Figure 4a). Moreover, the correlation between MDR and mean response provides a mechanism for higher brain regions to estimate the most informative future response changes of retinal ganglion cells based on the current response and thereby adjust the corresponding sensitivity, which could have benefits for the property of predictive coding [30]. Additionally, we found that

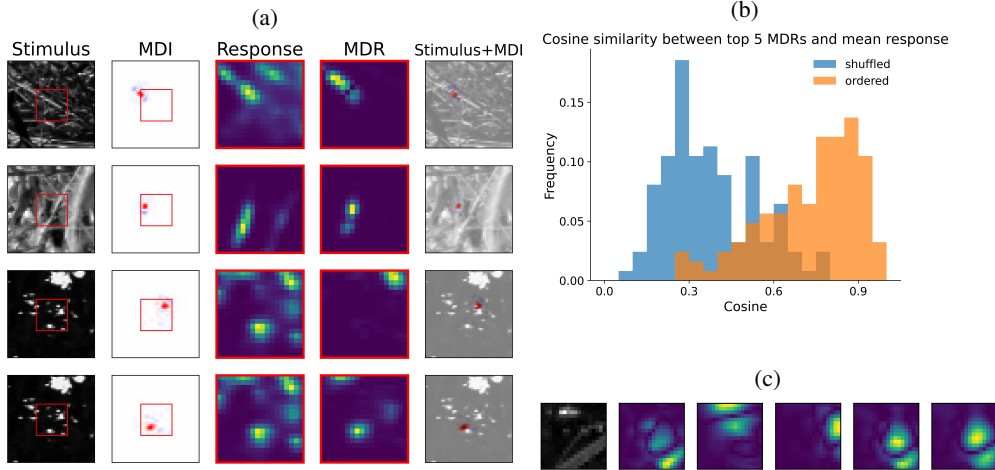

Figure 4: Most discriminable directions. (a) Examples of the most discriminable directions for different stimuli in the test set. The first column shows a representative frame of the spatiotemporal stimulus. The second column shows the spatial component of the spatiotemporal MDI obtained from the singular value decomposition. Red boxes correspond to the unit array in the final layer. The third column shows the mean response averaged over cell-type channels. The fourth column shows the absolute values of MDR averaged over channels. The last column shows the superposition of the MDI and stimulus. (b) Histogram of the cosine similarity between the mean response vector and its projection onto the space spanned by the top 5 MDRs. Similarities between shuffled pairs are also shown for baseline comparison. (c) One example of the stimulus (left panel, the central $18 \times 18$ array is shown) and the corresponding MDR for different cell type channels.

different parts of the stimulus generates the MDR for different cell types (Figure 4c), indicating that different cell types signal different stimulus regions in the image as conveying the most information.

### 4.3 Noise correlations in the retina limit information coding

The role of noise correlations in population coding has long been debated. Multiple scenarios have been advanced where noise correlations are detrimental to information coding [31, 10]. However, others suggest that noise correlations under certain circumstances may not limit information coding and can even be beneficial [32, 6, 9]. More specifically, it has been proposed that in response space, noise correlations decrease information when the differences in the signal are aligned with the noise direction, and increase information when they are orthogonal [33] (Figure 5a).

Here we tested these two hypotheses for the retina by computing cosine similarities between MDRs and MNRs. It turns out that the most discriminative mode is quite aligned with the most stochastic mode and orthogonal to modes with small variability (Figure 5b), implying that noise correlations in the retina are information-limiting correlations. We also performed a more direct test of this conclusion by analyzing trial-shuffled responses, which have independent noise. We found that removing noise correlations in this way created responses that were significantly more discriminative than the original model outputs with correlated noise for every population size (Figure 5c). To understand the mechanism of this reduction in information, we analyzed the eigenspectrum of the Fisher information matrix (Figure 5d). Both the sensitivity and the stochasticity increase with the discriminability of MDR, and their ratio reaches a maximum at the noisiest mode. Therefore, we propose that the fundamental reason for such information-limiting correlation is that noise propagates through the same network as the signal, causing the stochasticity ellipsoid to align closely with, and be rounder than, the sensitivity ellipsoid. One trivial and extreme example of this arrangement is when all noise originates at the level of the stimulus, e.g. from photoreceptors, then the resulting noise correlations will obviously be detrimental.

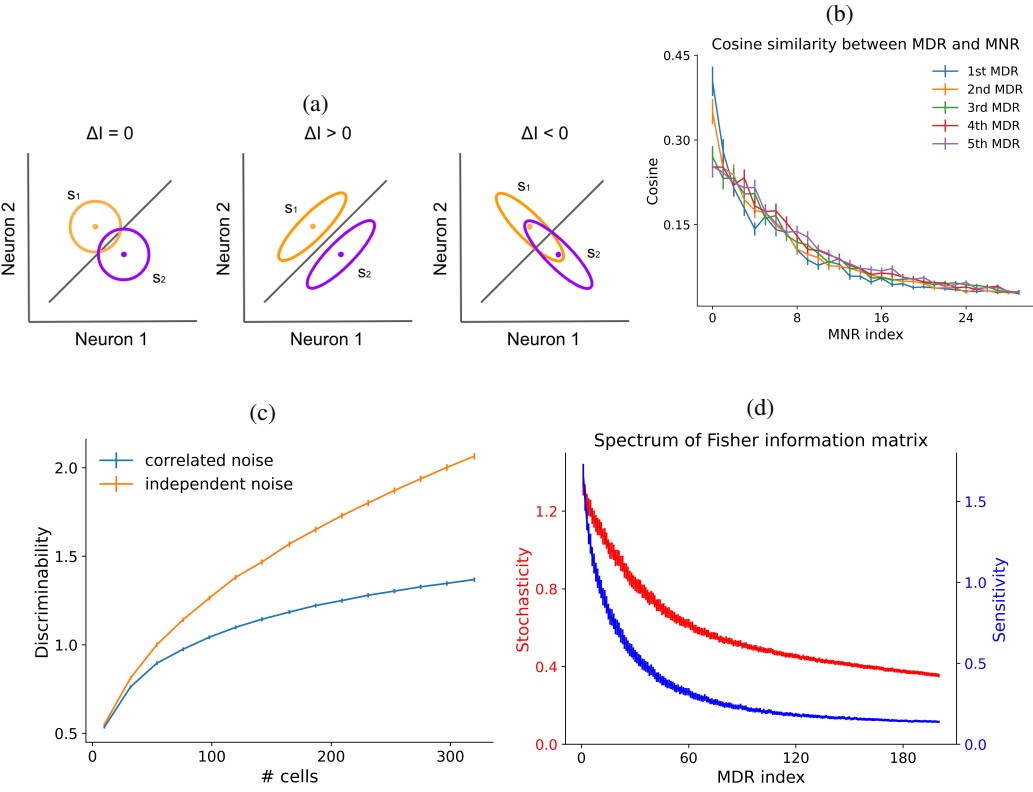

Figure 5: Noise correlations in the retina are detrimental to information coding. (a) Compared to independent noise (left), noise correlations can either increase information (middle) or decrease information (right) [33]. (b) Cosine similarities between the top 5 MDRs and the top 30 MNRs averaged over stimuli. (c) Discriminability of MDI for different numbers of output units. The plot for trial-shuffled responses with independent noise is shown for comparison. Output units are randomly selected from the central $500\mu m \times 500\mu m$ region [34]. (d) Sensitivity and stochasticity of the top 200 MDRs averaged over stimuli.

### 4.4 Firing rate dependency

The firing rate dependencies of stochasticity, sensitivity, and discriminability for individual neurons and the whole population are shown in Figure 6. For individual neurons, both the sensitivity and the stochasticity peak around intermediate firing rates and decay at high firing rates (a consequence of the refractory period). Thus, the single-cell discriminability also saturates around the intermediate firing rate. In contrast, population coding does not exhibit such a decay at high rates, implying that neurons can encode information complementarily, which is a benefit of population coding. In other words, even when a fraction of neurons get saturated by the stimulus and become uninformative at high firing rates, other neurons can still provide discriminative responses to stimulus changes. Although the population sensitivity and stochasticity significantly increase with the mean firing rate, the firing rate dependency of the population discriminability remains relatively flat, which is potentially beneficial for information transmission assuming that stimuli are uniformly distributed on the natural scene manifold [35, 36].

## 5 Discussion

In summary, we established a novel information-geometric framework to understand the high dimensional representation manifold of the retinal population for natural scenes, one that can also be

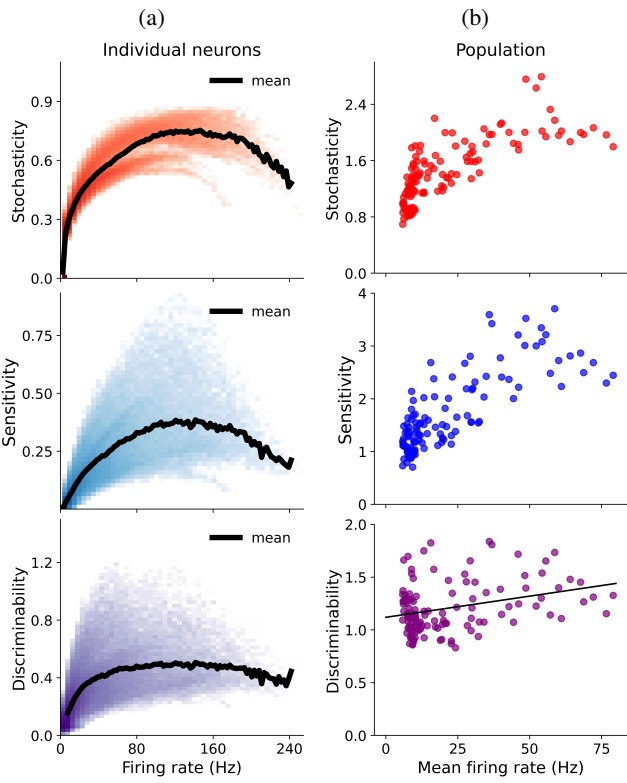

Figure 6: Firing rate dependency. (a) Log-scale 2D histograms showing the firing rate dependencies of stochasticity, sensitivity, and discriminability for individual neurons whose definitions are given in the theory section. Mean values were plotted with thick black lines. (b) Scatter plots showing the population-averaged firing rate dependencies of stochasticity, sensitivity, and discriminability for the first MDR. The linear regression of the populational discriminability is shown.

generalized to other neural systems. However, there still exist remaining questions and further work that can be pursued within this framework.

First, our analysis primarily focused on examining the local metric tensor, which characterizes the effects of infinitesimal stimulus changes. It would be interesting to extend the analysis to finite stimulus changes to explore more global geometry of the representation manifold. For example, one exciting direction is to find geodesics between two stimuli under the information metric and observe how one stimulus transitions into another with the least amount of discriminable changes for the retinal population [37].

Second, we found that noise correlations in the retina are information-limiting correlations. This conclusion is surprisingly robust against many hyperparameter selections, even when second order statistics are not fitted well. Therefore, it remains unclear whether this is a general result for all feedforward networks with independent noise applied to their inputs and intermediate layers. If this is indeed the case, noise correlations in other brain areas that do not limit information coding must arise from different mechanisms.

Finally, the Fisher information metric in this work was computed in the full stimulus space. However, most points in the stimulus space represent meaningless noise images that no animal has ever encountered. Therefore, an important avenue for future research would involve incorporating the structure of natural stimuli and computing the metric with stimuli constrained to a natural-scene manifold.

## Acknowledgements

This work was supported by grants from the NEI, R01EY022933, R01EY025087 and P30EY026877 (SAB).

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
