# Supplementary Material

## Information Geometry of the Retinal Representation Manifold

**Xuehao Ding**[1], **Dongsoo Lee**[2], **Joshua B. Melander**[3], **George Sivulka**[4], **Surya Ganguli**[5], and **Stephen A. Baccus**[6]

[1,5]Department of Applied Physics, Stanford University
[2,3]Neurosciences Phd Program, Stanford University
[4]Department of Electrical Engineering, Stanford University
[6]Department of Neurobiology, Stanford University
*{xhding, dsnl, melander, gsivulka, sganguli, baccus}@stanford.edu*

## 1 Retinal recordings and data preparation

A video monitor projected visual stimuli at 30 Hz controlled by Matlab (Mathworks), using Psychophysics Toolbox [1, 2]. Stimuli had a constant mean intensity of $8.3 \ mW/m^2$. Images were presented in a $50 \times 50$ grid with a square size of $55 \ \mu m$ on the retina. We presented a natural scene stimulus that was a sequence of jittered images sampled from a natural image database [3].

The responses of tiger salamander retinal ganglion cells from 5 animals were recorded using a 60 channel multielectrode array. Further experimental details are described in Ref. [4].

Spiking responses were binned using 10 ms bins and for the training set the responses were further smoothed using a 10 ms Gaussian filter. Each spatiotemporal stimulus spanned over 400 ms corresponding to the retinal integration timescale. For each preparation, the training dataset of 60 minutes in total was divided according to a $90\%/10\%$ train/validation split, and the test dataset consisted of repeated trials to novel stimuli of 60 seconds natural scene.

## 2 Hierarchical clustering of output channels

In order to determine the number of channels in the last convolutional layer and the neuron-to-channel map $C(k)$, we first fit a model for each preparation assigning a separate channel to each neuron, i.e., $C(k) = k$. We then computed the cosine similarity between the response vectors of every pair of different channels averaged over stimuli. A standard hierarchical clustering was then applied using the cosine similarity matrix as the affinity matrix. Neurons that were clustered into the same group then shared the same cell-type channel in the final model. The number of clusters as a hyperparameter was explored and the lowest number that did not limit the model performance significantly was chosen.

## 3 Binomial noise

We attempted a variety of probability mass functions (PMF) for modeling the sub-Poisson statistics of retinal ganglion cells. Apart from the binomial noise formulated in the main text, we also evaluated the parametric Gaussian noise used in Ref. [5] and two kinds of parametric Poissonian noise, as formulated by the following equations:

$$p^G(n; r, k, N) \propto \exp \left[ -\frac{(n-r)^2}{2k^2} \right], \quad n = 0, 1, ..., N \tag{1}$$

37th Conference on Neural Information Processing Systems (NeurIPS 2023).

$$p^{P1}(n'; r, k, N) \propto e^{-k \cdot r} \frac{(k \cdot r)^{k \cdot n'}}{(k \cdot n')!}, \quad k \cdot n' = 0, 1, ..., \lfloor k \cdot (N+1) \rfloor$$

$$p^{P1}(n|n') = \begin{cases} n+1-n' & n = \lfloor n' \rfloor \\ n'-n+1 & n = \lceil n' \rceil, \quad n = 0, 1, ..., N \\ 0 & \text{otherwise} \end{cases} \tag{2}$$

$$p^{P2}(n; r, k, N) \propto e^{-k \cdot r} \frac{(k \cdot r)^{k \cdot n}}{\Gamma(k \cdot n + 1)}, \quad n = 0, 1, ..., N \tag{3}$$

For each PMF, we optimized the variability parameter $k$ to maximize the log-likelihood of empirical spike counts based on empirical firing rates. We then computed the KL divergence between each PMF and the empirical spike count distribution at different firing rates. As shown in Figure 1, the binomial noise and the second type of Poissonian noise performed the best, and we ultimately chose binomial noise because of its interpretability.

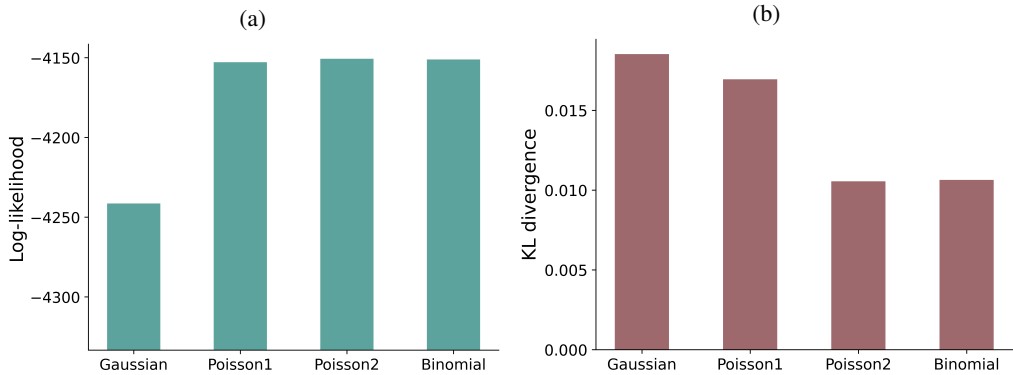

Figure 1: (a) The log-likelihood of empirical data for each PMF averaged over cells. (b) The KL divergence between model prediction and empirical data for each PMF averaged over cells.

We compared the variance-mean relation between the empirical data and the binomial PMF with the optimized parameter $k$. Example cells are shown in Figure 2. Binomial noise captures the empirical relation very well except for some outliers that fluctuate more because of their small sample sizes.

## 4 Statistical measures

Let the neural response be represented by the tensor $R_{tsc} \in \mathbb{N}$ denoting the spike count of the $c$-th cell during the $s$-th time bin in the $t$-th trial. Statistical measures used in the main text are defined as follows.

Total pairwise correlation:

$$C_{i,j}^{total} := \langle \frac{\langle (R_{tsi} - \langle R_{ts'i} \rangle_{s'}) \cdot (R_{tsj} - \langle R_{ts'j} \rangle_{s'}) \rangle_s}{\sqrt{\langle (R_{tsi} - \langle R_{ts'i} \rangle_{s'})^2 \rangle_s \cdot \langle (R_{tsj} - \langle R_{ts'j} \rangle_{s'})^2 \rangle_s}} \rangle_t. \tag{4}$$

Stimulus correlation:

$$C_{i,j}^{stim} := \frac{\langle (\langle R_{tsi} \rangle_t - \langle R_{ts'i} \rangle_{ts'}) \cdot (\langle R_{tsj} \rangle_t - \langle R_{ts'j} \rangle_{ts'}) \rangle_s}{\sqrt{\langle (R_{tsi} - \langle R_{t's'i} \rangle_{t's'})^2 \rangle_{ts} \cdot \langle (R_{tsj} - \langle R_{t's'j} \rangle_{t's'})^2 \rangle_{ts}}}. \tag{5}$$

Noise correlation:

$$C_{i,j}^{noise} := \frac{\langle (R_{tsi} - \langle R_{t'si} \rangle_{t'}) \cdot (R_{tsj} - \langle R_{t'sj} \rangle_{t'}) \rangle_{ts}}{\sqrt{\langle (R_{tsi} - \langle R_{t's'i} \rangle_{t's'})^2 \rangle_{ts} \cdot \langle (R_{tsj} - \langle R_{t's'j} \rangle_{t's'})^2 \rangle_{ts}}}. \tag{6}$$

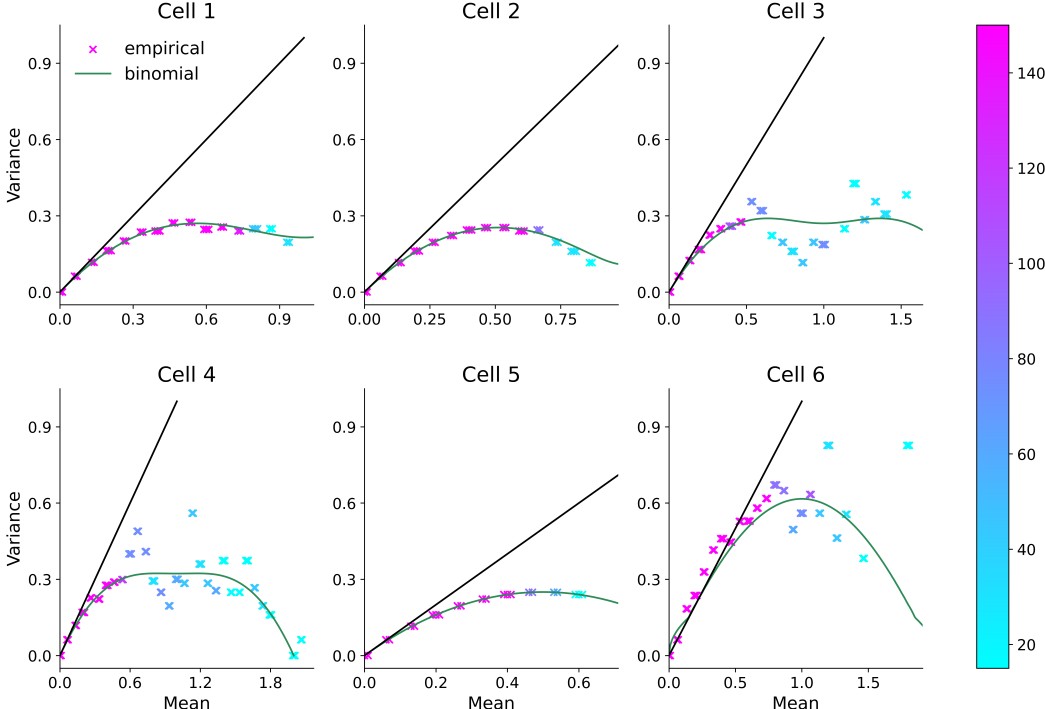

Figure 2: Variance-mean plots for empirical spike counts and the binomial PMF after optimization. Each empirical point represents the statistics of samples in all time bins with similar firing rates and its color codes the sample size. Black line is the identity line.

Trial-to-trial correlation:

$$C_i^{trial} := \langle \frac{\langle (R_{asi} - \langle R_{as'i} \rangle_{s'}) \cdot (R_{bsi} - \langle R_{bs'i} \rangle_{s'}) \rangle_s}{\sqrt{\langle (R_{asi} - \langle R_{as'i} \rangle_{s'})^2 \rangle_s \cdot \langle (R_{bsi} - \langle R_{bs'i} \rangle_{s'})^2 \rangle_s}} \rangle_{a \neq b}. \tag{7}$$

Fano factor:

$$F_i := \langle \frac{\langle (R_{tsi} - \langle R_{t'si} \rangle_{t'})^2 \rangle_t}{\langle R_{tsi} \rangle_t} \rangle_s. \tag{8}$$

In some previous work, the stimulus correlation is alternatively defined as the average correlation between the neural responses of two cells in different trials. Correspondingly, the noise correlation is defined as the difference between the total correlation and the stimulus correlation. In practice, this set of definitions always produced very similar values to our definitions.

## 5  Examples of the most discriminable stimulus direction

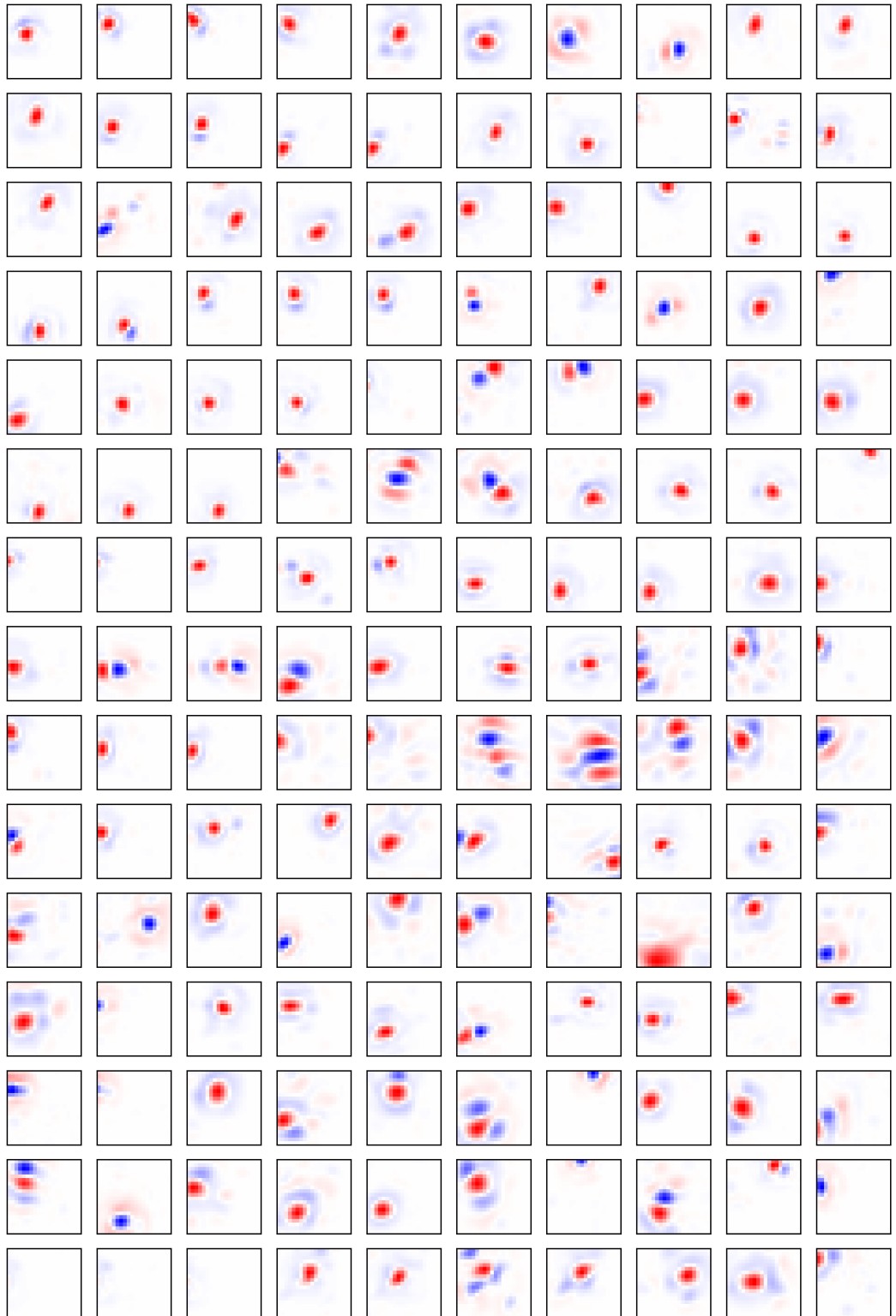

Figure 3: More examples of the spatial component of the most discriminable input (MDI) for different stimuli. The central $20 \times 20$ arrays are shown.

# 6 Examples of the least discriminable stimulus direction (LDI)

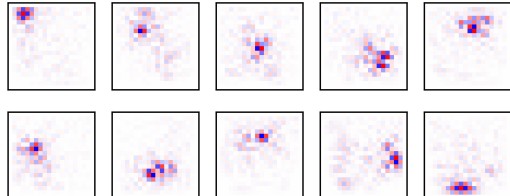

Figure 4: Examples of LDI (spatial component) across stimuli. For the purpose of fair comparison, here we only allow changes of the central pixels that can affect an equal (and maximal) number of output units. The central $20 \times 20$ arrays are shown.

# 7 Relations among individual sensitivity, stochasticity, and discriminability

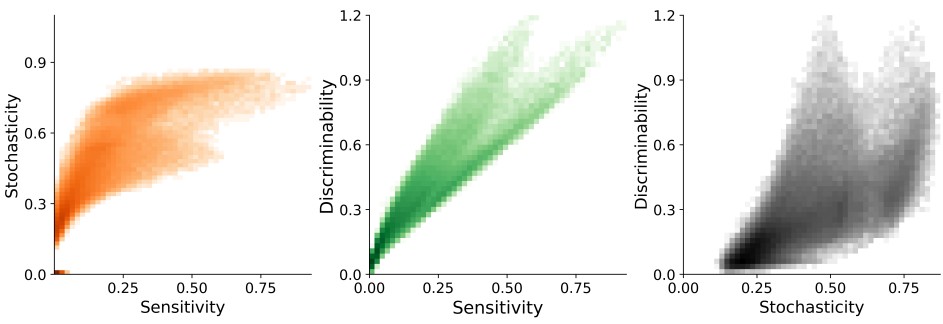

Figure 5: Log-scale 2D histograms showing the relations among individual sensitivity, stochasticity, and discriminability across a large group of stimuli.