# OpenReview forum: "Information Geometry of the Retinal Representation Manifold"
_NeurIPS.cc/2023/Conference — NeurIPS 2023 poster_

### Official Review · Reviewer_pZyp · 2023-06-23

**Soundness:** 3 good
**Presentation:** 3 good
**Contribution:** 3 good
**Rating:** 6
**Confidence:** 4

**Summary:**

This paper uses a fitted surrogate neural network model to approximate the Fisher information metric induced on image space by the retinal ganglion cell population code. They use this analysis to argue that noise correlations in the retina are information-limiting.

**Strengths:**

1. The question of whether noise correlations in early sensory areas are information-limiting is a classic and broadly interesting issue in theoretical and systems neuroscience. This paper offers a novel approach to the problem of Fisher information estimation.

2. I think the observation that the most discriminable stimulus depends on the base point (Lines 196-208) is an intriguing finding, though perhaps not so surprising. Making a convincing link between this variation and neural adaptation would be an interesting topic for future study (I fully acknowledge that a detailed characterization of this phenomenon is likely beyond the scope of the present manuscript).

**Weaknesses:**

1. I have several concerns regarding the robustness of the paper's conclusions to the architecture and goodness of fit of the surrogate model. One key strength of some past work on information limiting correlations---I have in mind Rumyantsev et al, cited as [8] in the submitted manuscript---is the demonstration that the quantities of interest can be accurately resolved given a number of measurements comparable to the number of experimental recordings. It is not clear to me whether the same should be true here. Can the authors provide evidence that their approach gives accurate estimates of the directions of maximal variation, and that the surrogate model is not overfit?

2. In a similar vein, the authors argue reasonably convincingly that it is reasonable to neglect stimulus-dependence of the noise covariance (i.e., $d\Sigma/dx \simeq 0$) for a ReLU network model, but do not give direct evidence that this is a reasonable assumption for the retinal population code.

3. In several places, data supporting the authors' analysis decisions are not shown, and their choices are not always clearly described. For example, in the truncated approximation (8) for the Fisher information matrix, can you specify (at least in the SI) the precise criterion used to select the "hundreds of the most stochastic modes" included (Lines 170-171)?

**A pedantic concern:** The paper and supplementary material contain several small violations of the anonymity requirements. My score for the paper does not take this concern into account.

- Lines 257-259: "Acknowledgements: This work was supported by grants from the NEI, R01EY022933, R01EY025087 and P30EY026877 (SAB)."

- In the supplementary ZIP, under code, the LICENSE file contains the line "Copyright (c) 2023 Baccus Lab."

**Questions:**

- Line 33: In addition to Wang & Ponce 2021, earlier work by Shao et al., "The Riemannian geometry of deep generative models" (2018) should be cited.

- Lines 73-81: It could be useful to mention that (8) has been termed the "linear Fisher information" in past works, and to cite Beck et al., "Insights from a Simple Expression for Linear Fisher Information in a Recurrently Connected Population of Spiking Neurons" (NECO 2011) and Kanitscheider et al., "Measuring Fisher Information Accurately in Correlated Neural Populations" (PCBI 2015).

- Line 84: Why is [1] cited rather than a general text on Riemannian manifolds (or, indeed, [14])?

- Line 86: Using the acronym "MDS" for "most discriminable stimulus" conflicts with the standard use of "MDS" to mean "multidimensional scaling." Please consider using an alternative acronym, e.g., "MDI" for "most discriminable input."

- Lines 152-153: Though the binomial noise model yields the best fit, could it still be worthwhile to reproduce Figure 5 for the alternative noise models, as a sort of robustness check?

- Line 157: Please provide more detailed information on compute resources than "on NVIDIA GPUs."

- Lines 185-186: "Statistics for preparations with longer test sequences tend to be more reliable and consequently our model performs better on these." Data to support this claim is not shown, correct? It would be useful to show more clearly the effects of this heteroskedasticity.

- Figure 4: Please state in the caption whether these measures are computed on held-out stimuli.

- Lines 206-208: Could you make this apparent link to ideas of hierarchical predictive coding more precise?

- Figure 6a-b: It would be useful to remind the reader of the definitions of stochasticity, sensitivity, and discriminability in the caption.

- Figure 6a: In the three sub-panels of Panel (a), there are so many dots overlaid on top of each other that it is hard to tell how single neurons are distributed within the blobs of data. Showing 2D histograms might be more informative. It would also be useful to plot stochasticity, sensitivity, and discriminability against one another rather than only against firing rate (depending on the result, this could be deferred to the supplement).

- Figure 6b: The linear fit to the discriminability-mean firing rate relationship is not very convincing because of the substantial spread in the data.

- Lines 244-247: I agree that it would be interesting to compute geodesics, but I'm less optimistic that this could be done in practice due to the numerical challenges associated with solving the geodesic equation in high dimensions. One previous attempt by Hénaff and Simoncelli ("Geodesics of learned representations," ICLR 2016) used an optimization-based approach, with somewhat mixed success. Can you elaborate on why you think this is a realistic possibility?

**Limitations:**

The authors provide some discussion of the limitations of their work, but I think a more comprehensive assessment of the possible failings of each step of their approach would enhance the paper.

---

> ### Author Rebuttal · Authors · 2023-08-04
>
> Thanks for your careful review and helpful suggestions.
>
> Weaknesses:
>
> 1. We are analytically computing the Fisher information given an excellent model of the retina. Rumyantsev et al. in contrast had to estimate Fisher information directly from data without a model. Therefore they had to worry about whether they had enough trials and neurons. We don't have to worry about this because our model is accurate to two essential properties under natural scenes, the sensitivity as captured by an accurate model of the neural code, and an accurate fit of noise correlations. Given these measures, everything afterwards in computing Fisher information is analytic.
>
> 2. $\frac{d\Sigma}{dx}$ is expected zero for any point in stimulus space that yields mean hidden neuron activities where every ReLU neuron's activation is a reasonable distance away from a zero-crossing, measured in units of the standard deviation of noise injected into the ReLU.   Therefore, for the model, this statement is true over most of stimulus space except for those stimuli which place  the mean response of one or more ReLU's close to their zero crossing.  In essence, whenever every ReLU's mean response is just greater than its own noise input's standard deviation away from its zero crossing, small changes in stimulus do not change the linear response of the overall model, and therefore do not change the noise correlations $\Sigma$.  Now given our model is an excellent model of the retina itself, this statement should be a good approximation for the retina.  One could test this by providing two nearby stimuli to the retina and directly measuring noise correlations and showing they do not change much.  But this new biological experiment is beyond the scope of this paper.  We hope our direct prediction of measured noise correlations suffices.
> 3. We usually consider the top 500 most stochastic modes (top 500 principal components) that can explain 85% to 90% of the total variance depending on the stimulus. Additionally, as shown in Fig. 5b, MDS mostly correlates with the top 30 most stochastic modes. Therefore, we claim that our summation converges for the purpose of computing MDS. We can state our criterion of this selection more explicitly in the final version.
>
> Questions:
>
> 1. Thank you for adding the new citation, we will cite it in the final version.
> 2. We will mention the terminology and add the citations in the final version.
> 3. We cited reference [1] as the source of the insight that a Riemannian metric can transform between stimulus space and representation space, whereas a general text typically doesn't concern neural networks or sensory systems.
> 4. Thanks for bringing this up, we will use MDI instead in the final paper.
> 5. Fig. 5 is about the geometry of noise correlations. That is to say, the result in Fig. 5 is a product of Gaussian noise added before the final layer. Noise added to the final layer, no matter it is binomial noise or other noise models, is independent noise, will not affect the conclusion in Fig. 5. To confirm this, we have plotted Fig. 5 even without the final independent noise, and the conclusion holds. We can clarify the role of the final independent noise.
> 6. We will provide the detailed GPU information in the final paper.
> 7. In the caption of Fig. 3, we stated that "the circle radius is proportional to the square root of the total length of the test set". And one can indeed see from Fig. 3 that larger circles are matched better.
> 8. The stimuli belong to the test set.
> 9. The correlation between the MDR and mean response provides a mechanism for a higher brain region to estimate the most informative response changes of retinal ganglion cells in the immediate future based on the current response. Using a strategy known as Bayesian Inference,  the higher brain by detecting the response, $R$, could potentially increase sensitivity to the most informative $\Delta R$ in order to extract the most salient and informative features of the stimulus, analogous to attentional cueing. In addition, by subtracting out the mean response to emphasize the informative $\Delta R$, the higher brain could use predictive coding, which yields a more efficient representation by encoding the prediction error. A combined optimal readout strategy may combine both elements, known as Bayesian Predictive Coding (Aitchison et al. 2017).
> 10. Yes, we can remind readers that the definitions are in the theory section.
> 11. We agree that the suggested plots directly comparing different quantities could be put in the supplementary material, since they are less relevant to our main conclusions but some readers might be interested.
> 12. Yes, actually we should have set the y-axis to start from 0 since we are looking at the fluctuation relative to the absolute value. In that case we believe that the plot will look much more convincing. (See Fig. A4 in the attached pdf.)
> 13. We also believe that it would be impractical to solve the high-dimensional differential equation to find the geodesic rigorously for our model. But there could exist approximation methods to help answer the question. The first step could be computing the geodesic distances of a finite set of paths. With these data it becomes possible to refine the path searching space if paths with small geodesic distances share any common features (dimensionality reduction methods could be used here). Applying some constraints to the path in the pixel space with Euclidean metric could accelerate the searching. Reducing the full model to some approximately equivalent small model would also help.

---

> > ### Comment · Reviewer_pZyp · 2023-08-11
> >
> > I thank the authors for their detailed response to my comments and those of the other reviewers. I think this is an interesting contribution, and I will raise my score.

---

### Official Review · Reviewer_SiWR · 2023-06-30

**Soundness:** 3 good
**Presentation:** 2 fair
**Contribution:** 2 fair
**Rating:** 5
**Confidence:** 3

**Summary:**

This paper uses a CNN-based model fit to recorded responses from salamander retinal ganglion cells to explore the Fisher information matrix $I$ of neural firing. They report that top eigenvectors of $I$ vary markedly with the stimulus being shown and that most discriminative response modes often align with top eigenvectors of $I$. They further argue, on the basis of their analysis, that noise correlations in the retina are likely to be information-limiting because signal and noise are propagated via feedforward mechanisms through the same channels.

This is an interesting paper that employs information geometry methods to provide some insights into retinal coding. However, the results are somewhat preliminary, and it is unclear how closely these insights are tied to a particular model architecture. The presentation of some aspects of model training is also somewhat confusing.

**Strengths:**

- Good fits to experimental data.
- Attempts to link models to known physiology. Well-grounded in current neuroscientific theories and questions.
- While Fisher information has been widely used to investigate neural population coding, the use of a model fit to natural image data to estimate Fisher information over a wider range of stimuli is innovative.

**Weaknesses:**

- Several modeling choices seem somewhat _ad hoc_, and it's unclear how much the results depend on them. For instance, numbers of hidden layers, types of nonlinearities, and structure of injected noise might possibly play a role, but it's unclear. The authors would likely argue that their good fits to summary statistics of data obviate these questions, since the role of the model is simply to interpolate between measured data for purposes of the manifold analysis, but that's to be shown.
- Section 3.2 details a somewhat surprising hodgepodge of optimization strategies for different components of the model, suggesting the results depend fairly sensitively on how training is done.
- Results are intriguing but seem somewhat preliminary. For instance, the analysis of MDS and MDR in Figure 4 is interesting, but I'm missing the bigger picture about what to take away from this.

**Questions:**

- Is there a reason the authors have chosen the non-standard binomial parameterization in Equation 7? Of course it is binomial with sufficient statistic $n$ and natural parameter $k\log \frac{r}{1-r}$, which would normally be written $\log \frac{p}{1-p}$, but calling $k$ a "variability parameter" seems somewhat misleading, as this is still a binomial distribution with a single parameter. It is stated that $N$ is cell-dependent. What about $r$ and $k$?

- I found the exposition of the one-hot layer in Section 3.1 very difficult to follow. I understand that the goal is to mimic retinal organization, but I'm still unclear how this matches up with the math in Eqs. 5 and 6. For instance, I don't know what `input` means in (5). Are $i$ and $j$ indices in the tensor of the last layer, e.g., `W[C(k), i, j]`? And $k$ indexes neurons? Are distinct $(i, j)$ supposed to correspond to distinct RF locations? Similarly, in (6), what do $(p, q)$ correspond to conceptually? It took me a while staring at the formula to understand why a 1-hot $\mathbf{W}$ is optimal for a given $p$ and $q$, and then the sum just takes care of all possible assignments? And I'm still not clear on how this one-hot assignment for each $k$ prevents output units with different $k$ from using the same $(C, i, j)$. I think a conceptual figure would be a real help here.

- Minor: Line 166: This $w$ is not the same as the one in (6), correct?

**Limitations:**

- As stated in the paper, the calculations are predicated on the assumption that the noise correlations $\boldsymbol{\Sigma}$ are locally independent of the stimulus. This may be optimistic in some cases.
- As noted above, the results are conditioned on a particular choice of model architecture, which may or may not affect the results.
- Since the model is based on fits to a limited number of ganglion cells in salamander, it may or may not generalize well to the entire retinal population or other species.

---

> ### Author Rebuttal · Authors · 2023-08-03
>
> Thank you very much for your review and helpful critiques.
>
> Weaknesses:
>
> 1, 2.
> Please see the overall author rebuttal as to choices of model architecture and optimization. Most importantly, we note that our neural architecture is building on previous work that accounted for the mean deterministic but not variable stochastic responses of the salamander retina to natural movies (see our cited [11,12,13]). In that work an extensive architecture search was performed which varied number of layers, number of channels per layer, etc... to find the simplest model that can capture mean ganglion cell responses.  We are building on this extensive architecture search by taking the SOTA model and showing, remarkably, that without any further modification, other than the injection of optimized independent noise in each layer, we can accurately capture many many pairwise noise correlations in the retinal ganglion cell output.  This in and of itself is a major contribution to computational neuroscience: nobody has ever accurately captured such retinal noise correlations, for natural scenes, using a feedforward neural circuit model, whose internal structure matches that of the biological retina.
>
> 3.
> We agree that many MDSs and MDRs are difficult to interpret visually. Thus, our main conclusions lie in Fig 5,6, particularly related to the ongoing question about the role of noise correlations as described in the overall author rebuttal. In addition, our recent research finds that when constrained on a natural scene manifold, MDS will become more visually interpretable.
>
> Questions:
>
> 1.
> The rationale for choosing the non-standard binomial distribution is the refractory period of retinal ganglion cells which induces a maximal spike count in a time bin. So $N$ is the cell-dependent parameter that we can directly determine from the observed refractory period in our experimental recording. After fixing $N$, $k$ is the parameter controlling the variability of the distribution (also cell-dependent), and $k$ can only be determined through model optimization. $r$ is not a parameter to be optimized, it rather controls the firing rate which is updated according to the current CNN output. We can improve our description about the binomial noise in the final paper to make it more clear.
>
> 2.
> The input to the one-hot layer is the output of the last convolutional layer, which is a 3d tensor where the first index is the channel index and the other two indices $i,j$ refer to the location. $k$ is the index of recorded neurons. So Eq.5 says that each output unit is a linear combination of units in different locations in a channel, and such combination weights will eventually converge to a one-hot vector after model training.
>
> In Eq. 6, $p,q$ are also location indices like $i,j$. To intuitively understand why a one-hot $w^k$ can minimize the loss function, the key observation is that, suppose $w^k$ is a one-hot vector, then $|\prod_{ij}(w^k_{ij}+\delta_{pq,ij}-1)|=1$ for $(p,q)$ such that $w^k_{pq}=1$. While if $w^k$ is far from a one-hot vector, then such product will produce an extremely small number for any $(p,q)$. We will describe this more clearly in the final paper.
>
> 3.
> Correct. We will use a different notation to denote the eigenvalue in the final paper.
>
> Limitations:
>
> 1. $\frac{d\Sigma}{dx}$ is expected zero for any point in stimulus space that yields mean hidden neuron activities where every ReLU neuron's activation is a reasonable distance away from a zero-crossing, measured in units of the standard deviation of noise injected into the ReLU.   Therefore, for the model, this statement is true over most of stimulus space except for those stimuli which place  the mean response of one or more ReLU's close to their zero crossing.  In essence, whenever every ReLU's mean response is just greater than its own noise input's standard deviation away from its zero crossing, small changes in stimulus do not change the linear response of the overall model, and therefore do not change the noise correlations $\Sigma$.  Now given our model is an excellent model of the retina itself, this statement should be a good approximation for the retina.  One could test this by providing two nearby stimuli to the retina and directly measuring noise correlations and showing they do not change much.  But this new biological experiment is beyond the scope of this paper.  We hope our direct prediction of measured noise correlations suffices.
>
> 2.
> See the overall author rebuttal as to the motivation of model architecture. In particular we are building on an extensive prior architecture search and our results are robust and reproducible across 4 different model fits to 4 different biological retinas.
>
> 3.
> It is a well-known anatomical principal that ganglion cells tile the retina, so at each location there is only one cell of a given type. This mosaic organization also occurs in the salamander retina (Fig. 2 in Kastner & Baccus, 2011) implying that single recorded neurons can be generalized by the one-hot layer to the population of the same cell type. It is acknowledged that all cell types may not have been recorded from. As to generalization to different species, the primary properties at play here, strong nonlinearity under natural scenes, the presence of noise correlations of a similar magnitude, similar nonlinear phenomenology and a mosaic ganglion cell organization have all been observed across different species of vertebrate retina.

---

> > ### Comment · Reviewer_SiWR · 2023-08-10
> >
> > I appreciate the authors' thoughtful responses. Replies to selected replies below:
> >
> > > 1, 2. Please see the overall author rebuttal as to choices of model architecture and optimization.
> >
> > Thanks for the clarification. This is helpful, but it should be informative to the authors that this point was made by three of four reviewers. Either additional pointers to the previous work or more discussion of modeling choices would not hurt.
> >
> > > $\frac{d\Sigma}{dx}$ is expected zero for any point in stimulus space that yields mean hidden neuron activities where every ReLU neuron's activation is a reasonable distance away from a zero-crossing, measured in units of the standard deviation of noise injected into the ReLU.
> >
> > Sorry, just to make sure I understand: this statement is predicated on the _assumption_ that the only factors affecting $\Sigma$ are how many neurons contribute, not where in stimulus space we are, correct? That is, once a neuron is into the linear portion of the ReLU, it is _assumed_ that the covariance is constant? I apologize if I'm missing something obvious here.

---

> > > ### Author Response · Authors · 2023-08-10
> > > **Constancy of Noise covariance is not an assumption when all ReLU's are far above or below threshold**
> > >
> > > To Reviewer SiWR: sorry we were not clear.   The local constancy of the noise covariance with respect to stimulus variation is not an assumption.  It can be proven to be true to good approximation when all ReLUs are either far enough above or far enough below threshold (i.e. 0 activation) in units of the input noise standard deviation.  Here is the proof.  First consider a linear network where the output $r$ is given by:
> > >
> > > $r = W (x + e^{in}) + e^{out}$.
> > >
> > > Here $x$ is the input stimulus vector, $e^{in}$ is an input noise vector, and $e^{out}$ is an output noise vector.  The noise covariance matrix $\Sigma^r$ of $r$ is straightforwardly computed to be:
> > >
> > > $\Sigma^r = W \Sigma^{in} W^T + \Sigma^{out}$,
> > >
> > > where $\Sigma^{in}$ and $\Sigma^{out}$ are the covariances of  $e^{in}$ and $e^{out}$ respectively.  This recovers the well known result that the output noise covariance $\Sigma^r$  of a linear network, conditioned on the input $x$, is actually completely independent of the input $x$.
> > >
> > > Now consider the nonlinear case:
> > >
> > > $r(x) = f(x + e^{in}) + e^{out}$.
> > >
> > > If the nonlinear map $f$ is a ReLU network where the input $x$ keeps the mean activity of each ReLU far from threshold (either above or below), in units of noise standard deviation, then over most of the noise distribution of  $e^{in}$ and $e^{out}$, we can Taylor expand $f(x + e^{in})$ about $e^{in}=0$, and the first order linear expansion will be a very good approximation.   Then the above result that the output noise covariance $\Sigma^r$ is independent of stimulus $x$ for a linear network applies to good approximation for the ReLU network: i.e.
> > >
> > > $\frac{d\Sigma^r}{dx} = 0$
> > >
> > > for the ReLU network. We hope this explains why  $\frac{d\Sigma^r}{dx} = 0$ is not an assumption but can be proven to be true to good approximation over almost all of stimulus space $x$.

---

> > > > ### Comment · Reviewer_SiWR · 2023-08-10
> > > >
> > > > Thanks for the clarification! I didn’t necessarily mean assumed prima facie but whether it followed from some assumptions in the model. This makes clear that it’s the linear + ReLU assumption that does it.

---

### Official Review · Reviewer_d8Ka · 2023-07-03

**Soundness:** 3 good
**Presentation:** 4 excellent
**Contribution:** 3 good
**Rating:** 6
**Confidence:** 4

**Summary:**

- This study uses an information geometry approach to studying visual coding in retinal populations.
- Using local-linear analyses (eigenmode + spectra) at different conditional responses of a CNN + fitted noise model, the authors observe
- The authors analyze the most sensitive coding directions of the population, as well as the co-alignment of the noise, and find that noise in the retina is information limiting.
- This is an interesting and conceptually clear/simple paper that adds to our understanding of neural coding in the retina.

UPDATE: Sep 1, 2023. I have read the rebuttal, it addressed my Qs and I already adjusted my confidence accordingly.

**Strengths:**

- I appreciate the clear writing style.
- The model and eigenvector analyses are conceptually simple to follow.
- The finding that noise correlations are information limiting is interesting.

**Weaknesses:**

- I am slightly concerned with the lack of effective null models against which to compare here. Given the size of the dimensionality. Either a positive or negative control. For example, is it conceivable that a different model with different noise eigenbasis, but with the same fit performance?

**Questions:**

- Similar Jacobian eigendecomposition analyses were conducted in Berardino et al. ("Eigen-distortions of hierarchical representations", Neurips 2017) to analyze model perceptual alignment with human visual perception. (In this paper they assume isotropic noise in the response domain, but in follow-up analyses in their dissertation, I believe they have more complex noise models.) Importantly, they not only consider the top eigenvector, but also the null space (eigenvectors with eigenvalue=0) as a means to probe model perception. I am wondering if such an analysis, systematically examining the null space of the model, would supplement your results with unique or complementary findings.
- Is there a difference in noise geometry for natural vs synthetic stimuli? Would these be captured by the model? I think a useful tool to quantify this would be recent work by Duong et al. ("Dissimilarity metric spaces for stochastic neural networks"; ICLR 2023), analyzing covariance orientation and scale for different classes of stimuli.
- Could there be a normative explanation for the noise and coding sensitivity co-alignment?
- Fig 5A should probably explicitly cite the Averbeck and Pouget review in the caption, saying that it is modified from there. Unless I am mistaken, it does look very similar.
- Minor: The abstract is a little long.

**Limitations:**

There was arguably some discussion of limitations but nothing explicit.

---

> ### Author Rebuttal · Authors · 2023-08-02
>
> Thank you very much for your review and helpful suggestions.
>
> Weaknesses:
> For the main result about the effect of noise correlation in the retina, we had already performed a null / control analysis that could be added to the final paper. As shown in Fig. A3 in the attached pdf, we shuffled trials of the model response to set all noise correlations to zero, thus creating a null model with independent noise and compared it with the original model. We found that the neural response with independent noise has significantly higher discriminability than the response with correlated noise, which further confirms our finding of how noise correlations reduce information under natural scenes.  We also note that our neural architecture is building on previous work that accounted for the mean deterministic but not variable stochastic responses of the salamander retina to natural movies (see our cited [11,12,13]).  There alternative architectures were considered, including the Generalized Linear Model (GLM), which was shown to perform poorly in modeling the mean response compared to the 3-layer CNN model (Maheswaranathan, Niru, et al. Neuron 2023).  Therefore we are extending a SOTA architecture and extending its applicability from mean responses to stochastic second order correlations in responses to natural movies, which in and of itself is a major contribution that has never been achieved before in the retina using a feedforward network whose internal hidden units and  computations match that of the biological retina.
>
> Questions:
> 1. Thank you for alerting us to this very relevant citation. We have also computed the least discriminable stimulus direction, and compared to the MDS which is more localized, they look more scattered and indeed difficult to discriminate for human eyes (Fig. A2 in the pdf). Since the least discriminable directions are less relevant to our main conclusions, they are not presented in the paper but we could add them in revision.
> 2. We believe that there is a large difference in the geometry for natural and synthetic stimuli. We also trained a model using checkerboard white noise stimuli and the resulting noise parameters and noise covariance geometry looks very different from that for natural scene, implying that noise in the retina highly depends on the overall stimulus statistics. Since the scope of our paper focuses on the much more ethologically important regime of natural scenes, we didn't present our results for white noise, but we can briefly describe describe such stimulus dependency.
> 3. Suppose that all Gaussian noise arises at the level of the stimulus, then obviously the noise and the signal directions will align. Deviation between noise and signal directions can only occur when the dominant noise source arises deeper in the network. In our case, a large fraction of noise source often arises earlier, implying that signal and noise will still be more aligned. Although we do not have a normative explanation as to whether the allocation of noise earlier in the network arises due to some advantage related to information or that it occurs due to mechanistic constraints (e.g.  photoreceptors having noise in the biochemical transduction cascade), we will improve our explanation of such co-alignment in the final paper. And as stated in the discussion section, one interesting future direction is to better understand under what circumstances such alignment would happen.
> 4. Yes, Fig. 5A  is modified from the Averbeck and Pouget review, cited in the text, and we will also add the citation in the caption in the final version of the paper.

---

> > ### Comment · Reviewer_d8Ka · 2023-08-11
> >
> > Thanks for your responses to my comments & Qs, and additional analyses.
> >
> > The model explaining retinal noise is already interesting by itself. However, I still believe additional work needs to be done (more controls) to dive into _why_ it works, and if this particular 2-step optimization procedure's fit could be produced by a different noise optimization procedure that may give a different conclusion. After reading other reviews, I believe what I'm getting at is related to other reviewers' comments on what feels like many ad hoc choices in this study, and the "hodge podge" of optimizations.
> >
> > That being said, I do like the main message of the paper (noise vs signal corrs of retina), and think it's a noteworthy result about an outstanding question in the field.  So I will maintain my score of 6, but am inclined to raise my confidence from 3 -> 4 to reaffirm that this would be of interest to others in the field.

---

> > > ### Author Response · Authors · 2023-08-13
> > >
> > > The reason for using a two-step optimization procedure is that the second order statistics are functions of the entire dataset and cannot be decomposed into terms depending on the individual stimulus. Thus it is not practical to optimize second order statistics along with mean firing rates. Fisher information requires an accurate measurement of both sensitivity and stochasticity, two properties that have never been optimized together before for natural visual scenes for any part of the nervous system. A highly accurate model of sensitivity has already been published after an extensive architecture search that included number of layers (1 - 3), number of channels (1 - 36), filter sizes and batch normalization. All of these deterministic properties were then completely fixed. Given the likelihood that optimizing sensitivity and stochasticity together might be intractable, it is amazing that the simplest possible procedure, namely fixing the deterministic sensitivity and just optimizing noise inputs for each layer (not even each channel separately) via grid search as well as the variability of the final independent spiking noise yields an accurate model of second order statistics. It didn’t have to be that simple - but it is remarkable that it was. Therefore our procedure to promote a model from capturing sensitivity to capturing discriminability, far from being ad hoc, is both simple and accurate - the best of both worlds. Given that stimulus discriminability under natural scenes is the real function of the visual system (not just sensitivity), our work is the first that shows that the true function of the retina can be modeled. The success of our method inspires confidence in two step optimization methods like this in future applications. As we stated in the overall rebuttal, for an accurate model of sensitivity and stochasticity, computing Fisher information is analytic. Therefore, although there may be other untested strategies for optimization, it is unlikely that there will be any solution of discriminability that differs from the current model.

---

> > > > ### Comment · Reviewer_d8Ka · 2023-08-14
> > > >
> > > > Thanks for your additional response.
> > > >
> > > > Indeed, I find value in your work and presented results. However, a stronger story should include a more thorough investigation of this noise model and other possible noise models. Therefore, I will be maintaining my score.

---

### Official Review · Reviewer_GBMw · 2023-07-10

**Soundness:** 3 good
**Presentation:** 2 fair
**Contribution:** 3 good
**Rating:** 5
**Confidence:** 4

**Summary:**

The authors present a new framework to understand stimulus discriminability for models of the visual pathway. They derive a Riemannian metric on the representation manifold and from there, different measures such as most discriminative/sensitive stimulus directions and their counterparts in the response space. The authors apply the method to a CNN model of the retina, fitted on experimental RGC data from tiger salamander. They find that discriminative stimulus directions are often aligned with stochastic modes and that populational codes can help in high firing rate regimes.

**Strengths:**

The authors present a framework which is quite general and can be applied to different models. This could even help to design better experiments and advance the geometrical understanding of neural representations.

**Weaknesses:**

While the theoretical framework is quite strong, the model presentation and the conclusions are not clear in some places. Especially the drawn conclustions are quite strong, and further investigation (for example (noise) model dependency) should be carried out.
More specifically I have the following remarks:
- Can the authors reiterate on the deduction of eq. 1 and how this defines a Riemannian metric on $\{ P(y|x)\}$?
- The model could be described more clearly.
- The optimization of the model is not completley clear to me and some clarification is needed:
  - how is the Poisson loss defined?
  - what is the exact (hierarchical) procedure?
  - maybe a short pseudo code would help here.

- The results seem to be highly dependent on the chosen model, and its noise assumptions. Firstly, it is not clear if they still hold if the model (or its noise) is changed. Secondly, the influence of the different noise assumptions could be tested. Thirdly, it is not clear how $g_i$ are exactly chosen (see optimization).

- While the model seems to be spatiotemporal, the authors purely focus on the analysis of the spatial component. What role does the temporal part play and can it be analyzed in the same framework?

**Questions:**

- How does the model compare to a Bayesian network, in which noise would be (a priori) independent (but systematically optimized) for each processing unit?
- How does the model perform on other performance measures? Even simple metrics such as reconstruction error, mean firing rate etc. could help to judge the results.
- How does the model compare to a simple baseline model (for ex. LNP)?
- Are the MDS robust across different models?
- Is there a way (or even meaningful) to compare the MDS to the RF of individual cells?
- How does the MDR (in Fig 4a) for individual cell type channels look like? Are they interesting and can be linked to RF?
- How do MDRs compare to other dimensionality reduction methods? For example, does it make sense to do PCA (or more fancy methods) on the neural data, push it forward to the stimulus space and compare to MDR?

**Limitations:**

Yes, the authors have addressed some limitations.

---

> ### Author Rebuttal · Authors · 2023-08-02
>
> Thank you very much for your review and helpful suggestions.
>
> Weaknesses:
> 1. The LHS of Eq. 1 is equal to $-2\int P(y|x)\ln[1+\frac{dP(y|x)}{P(y|x)}]dy$. We can add this line to help understand the derivation. The remaining step is just to apply a high-dimensional Taylor expansion to the logarithm function. Note that the first order term will vanish so only the second order term will be left. This is common knowledge in the information geometry (see citation [14)].
> 2. The model architecture is mainly described in the caption of Fig. 2 which is adopted from the architecture in citations [11, 12, 13] and more emphasis in the paper is given to the stochastic part. We can modify this part to make the description more clear and also emphasize more that this work follows from earlier studies.  Importantly, the model is a highly successful model that (1) underwent extensive architecture search to fit the mean response of salamander retina to natural movies; (2) it generalizes to correctly predict > a decade's worth of experiments on artificial stimuli; and (3) its hidden units and computations match that of the biological retina. We show remarkably that this *same* model could also correctly predict stochastic second order correlations.
> 3. Poisson loss is defined as loss(input,target)=input−target∗log(input)+log(target!). We used the regular Poisson loss in PyTorch. The detailed procedure of hierarchical clustering is summarized in the supplementary material. We computed the cosine similarity matrix across channels as the affinity matrix and applied the standard procedure with the function implemented in the scikit-learn package. We can define the Poisson loss and improve the description of hierarchical clustering in the final paper.
> 4. For the first two points, please see the overall rebuttal above. For the third point, we grid-searched $g_i$ to optimize the mean squared errors of noise correlations and stimulus correlations (weighted average), and more details of this can be provided.
> 5. It is true that the MDS is spatiotemporal. Analyses in Fig. 4b, Fig. 5, Fig. 6 are conducted with the full spatiotemporal MDS vector. We also computed the temporal components of MDSs but found that the temporal components for different stimuli are very similar, and are also very similar to the temporal components of instantaneous receptive fields. We did not find interesting results in terms of pure temporal components and therefore did not present them in figures considering the page limit, but can briefly describe these temporal components.
>
> Questions:
> 1. A Bayesian network is a probabilistic graph model, whereas our model is a mechanistically interpretable neural network whose internal units have correspondence with bipolar and amacrine cells as shown in citation [12] (Maheswaranathan et al., 2023). Rather than taking a different approach that would be disconnected with the biological implementation, in this manner it is important that the full sensitivity and stochasticity can be captured by taking a deterministic model and adding appropriate noise. The linearity and non-linearity in the model correspond to synaptic weighting / current integration and firing / vesicle release, respectively. We can emphasize more about the interpretability and mechanistic correspondence of our model in the final paper.
> 2. We computed the Pearson correlation between model firing rates and recorded firing rates, which ranges from 70% to 80%, which is the state-of-the-art performance for natural scenes. This is similar to the firing rate performance reported in citations [11, 12] because we are using a similar model architecture, which we can report it in the final paper.
> 3. The Linear-Nonlinear model and Generalized Linear Model (GLM), performs poorly under natural scenes compared to a 3-layer CNN model (Maheswaranathan, Niru, et al. Neuron 2023). Although GLMs have been shown to capture noise correlations under white noise stimuli (Pillow et al., 2008), the Linear-Nonlinear-Poisson model by definition cannot capture noise correlations between cells because noise is independent, which was tested explicitly in Meytlis et al. 2012. Such noise correlations are known to be important for decoding of stimuli under white noise and natural scenes (Ruda, Kiersten, et al. 2020).
> 4. Our experiments used four animals and we trained one model for each preparation. As stated more fully in the general rebuttal, comparing the results of these models, we believe that our results and conclusions are robust.
> 5. Since the receptive field is computed by averaging over stimuli, we think it is more meaningful to compare the MDS with instantaneous receptive fields (gradient of neural response with respect to stimulus), which is dependent on stimulus. According to the theorem we proved in the paper, the MDS lies in the subspace spanned by the instantaneous receptive fields of the output neurons. Indeed they often look similar, and sometimes the MDS looks like a superposition of two or three instantaneous receptive fields.
> 6. One example of MDR across different cell type channels as well as the stimulus is shown in Fig. A1 in the attached pdf. One can see that different parts of the stimulus generates the MDR for different cell types, indicating that different cell types signal different stimulus regions in the image as conveying the most information. We will add this figure to the paper and discuss briefly the cell-type dependency.
> 7. The most noisy response directions (MNR) are simply found by PCA with the result shown in Fig. 5b that MNR and MDR are correlated. We have not tried more sophisticated dimensionality reduction methods, which would be appropriate in the in the future to better understand the full structure of the neural manifold, but the relationship between MNR and MDR is highly likely to hold. We can emphasize the equivalence between PCA and the identification of the MNR in the final paper to help understand our method.

---

> > ### Comment · Reviewer_GBMw · 2023-08-14
> > **Re: Thanks for the clarifications.**
> >
> > I thank the authors for their thorough responses and I appreciate the additional explanations.
> >
> > While the similarities of the CNN to previous work were not obvious to me before (and appropriate explanations & references were missing), the model part becomes much clearer.
> > The authors highlight the mechanistic interpretability of their model, which one could argue about, compared to a detailed Hodgkin-Huxley-like model. But while this is not my intention here, a better explanation of the noise-model decisions would be beneficial, and if possible, how they can be interpreted mechanistically/biophysically (others than “synaptic noise” or “refractory period”), as the authors highlight this aspect.

---

> > > ### Author Response · Authors · 2023-08-15
> > >
> > > Thank you for the suggestion, it is a good point to discuss the mechanistic connection of the stochastic part of the model, because they are somewhat different in functional implication from the deterministic aspect discussed in previous publications. The deterministic portion was meant to capture the circuit architecture, thresholds at the presynaptic terminal and synaptic weighting of the retina in order to ascribe the functional effects of visual computations to individual cell types and synaptic connections, thus avoiding many details present in HH type models. However, when considering the stochastic model, other mechanisms may become important, and so it is worth drawing these connections.
> > >
> > > Gaussian noise added in our model is prior to the threshold, which mechanistically would correspond to voltage-dependent calcium channels in the synaptic terminal. In the first layer, noise will be dominated by the phototransduction cascade (Ala-Laurila & Rieke, 2011) as well as ion channel noise that when combined through addition of membrane currents will tend to a Gaussian distribution in the bipolar cell membrane potential. In the second layer, noise from bipolar cell vesicle release (after the first layer threshold) and ion channel noise in amacrine cells will similarly tend towards Gaussian. The non-monotonic (inverted U-shaped) single cell noise in ganglion cell spiking classically appears in a variance - mean plot for binomial processes such as voltage-dependent ion channel gating, receptor activation or spiking that exceed p > 0.5.
> > >
> > > We feel it is an interesting conclusion that although there may be a number of biophysical noise sources that differ microscopically from a Gaussian distribution (e.g. Poisson noise in vesicle release), when summed through the circuitry determined by the deterministic model, Gaussian noise becomes the best model, outperforming Poisson noise. Thus the central limit theorem simplifies the model’s optimization.

---

### Author Rebuttal · Authors · 2023-08-05

Thank you for pointing out the strengths of our work. As stated by two reviewers, a primary important contribution of our work is that we have solved a long-lasting debate about the role of noise correlations in the retina under natural visual stimuli using a novel information-geometric framework, one that can also be generalized to other neural systems. More generally, although numerous analyses have been conducted about discriminability for simple stimuli, and deterministic sensitivity under natural scenes have been modeled, our stochastic model is the first to be able to capture both sensitivity and stochasticity under natural stimuli for any system, thus enabling a broad set of questions about stimulus discriminability that were previously inaccessible.

One shared concern of the reviewers is the rationale of our selections of the model, hyperparameters, and optimization method and whether the scientific results are robust against these selections. Although these are reasonable concerns, some reviewers may not be familiar with previous work that is the foundation of our current model, which consists of the deterministic part (the CNN and the one-hot layer) and the stochastic part (Gaussian and binomial noise). We will discuss these two parts separately.

The CNN architecture and corresponding hyperparameters are adopted from citations [11, 13] and (Maheswaranathan, et al. 2023), including the number of layers, the number of channels, nonlinearities, implementation of convolutional layers, etc.. This previous work has not only shown that the current model setting can achieve the state-of-the-art performance in fitting the RGC mean firing rates for natural scenes, but also that the model has a clear correspondence with the real retina given the phenomenology reproduced by the model and the correlation between the activity of model internal units and recorded interneuron responses that the model was never fit to. Therefore, this CNN is not simply a statistical approach for fitting experimental data, but rather a mechanistically interpretable retinal model that captures the computations, circuits and representations of the salamander retina.

For the stochastic part of the model, which is new to our current work, it is important to note that our optimization did not change the deterministic parameters described above, but only optimized the addition of noise parameters to capture as much as possible the full properties of second-order retinal stochasticity as a function of stimulus and response. We have attempted a variety of approaches in fitting these second order statistics, including Poisson noise in intermediate layers, using Gaussian noise in the final layer, neglecting the refractory period, only fitting noise correlations, and so on. We have concluded that the current stochastic setting and optimization methods can fit second order statistics most accurately and fit different experimental preparations robustly. We understand that our optimization method for different model components may look complicated, but this is because fitting multiple second order statistics simultaneously across a large natural stimulus set is non-trivial and challenging.

To test robustness, we have tested models trained for four experimental preparations with different CNN and noise parameters, different number of output channels and cell-to-channel mappings,  different types of nonlinearities and different implementations of the convolutional layer, and different noise parameter selection criterions. We claim that all results reported in the paper are robust against these variations.

Furthermore, the Fisher information matrix only depends on sensitivity and the noise covariance matrix (stochasticity). Sensitivity was computed with the noiseless model which has been validated extensively by previous works ([11, 13] and Maheswaranathan et al. 2023), whereas the covariance matrix is proportional to the noise correlation matrix, which is captured well by our model as shown in Fig. 3A. Therefore, we claim that our computation of Fisher information metric is reliable.

In the final paper, we will emphasize more about the relation between our work and previous literatures, the robustness of our results, and the challenge of fitting experimental data.

---

> ### Author Response · Authors · 2023-08-13
>
> The rationale for our optimization procedure has come up several times so we thought we would repost our response to reviewer d8Ka again here.
>
> The reason for using a two-step optimization procedure is that the second order statistics are functions of the entire dataset and cannot be decomposed into terms depending on the individual stimulus. Thus it is not practical to optimize second order statistics along with mean firing rates. Fisher information requires an accurate measurement of both sensitivity and stochasticity, two properties that have never been optimized together before for natural visual scenes for any part of the nervous system. A highly accurate model of sensitivity has already been published after an extensive architecture search that included number of layers (1 - 3), number of channels (1 - 36), filter sizes and batch normalization. All of these deterministic properties were then completely fixed. Given the likelihood that optimizing sensitivity and stochasticity together might be intractable, it is amazing that the simplest possible procedure, namely fixing the deterministic sensitivity and just optimizing noise inputs for each layer (not even each channel separately) via grid search as well as the variability of the final independent spiking noise yields an accurate model of second order statistics. It didn’t have to be that simple - but it is remarkable that it was. Therefore our procedure to promote a model from capturing sensitivity to capturing discriminability, far from being ad hoc, is both simple and accurate - the best of both worlds. Given that stimulus discriminability under natural scenes is the real function of the visual system (not just sensitivity), our work is the first that shows that the true function of the retina can be modeled. The success of our method inspires confidence in two step optimization methods like this in future applications. As we stated in the overall rebuttal, for an accurate model of sensitivity and stochasticity, computing Fisher information is analytic. Therefore, although there may be other untested strategies for optimization, it is unlikely that there will be any solution of discriminability that differs from the current model.

---

### Decision · Program_Chairs · 2023-09-21

**Decision:**

Accept (poster)

**Comment:**

This paper performs a novel analysis of neural responses in the salamander retina.  They find that natural scene noise correlations limit information transmission which is opposite to a common speculation.   Reviewers found the initial paper did not clearly justify model design choices and properly relate the work to the prior literature, but upon rebuttal and discussion, reviewers all agreed that the paper was worthy of acceptance providing that these clarifications are added to the paper.